# STree: Speculative Tree Decoding for Hybrid State-Space Models

**Yangchao Wu**[1][*]  **Zongyue Qin**[1]  **Alex Wong**[2]  **Stefano Soatto**[1]

[1]**UCLA**  [2]**Yale University**

## Abstract

Speculative decoding is a technique to leverage hardware concurrency in order to enable multiple steps of token generation in a single forward pass, thus improving the efficiency of large-scale autoregressive (AR) Transformer models. State-space models (SSMs) are already more efficient than AR Transformers, since their state summarizes all past data with no need to cache or re-process tokens in the sliding window context. However, their state can also comprise thousands of tokens; so, speculative decoding has recently been extended to SSMs. Existing approaches, however, do not leverage the tree-based verification methods, since current SSMs lack the means to compute a token tree efficiently. We propose the first scalable algorithm to perform tree-based speculative decoding in state-space models (SSMs) and hybrid architectures of SSMs and Transformer layers. We exploit the structure of accumulated state transition matrices to facilitate tree-based speculative decoding with minimal overhead relative to current SSM implementations. Along with the algorithm, we describe a hardware-aware implementation that improves naive application of AR Transformer tree-based speculative decoding methods to SSMs. Furthermore, we outperform vanilla speculative decoding with SSMs even with a baseline drafting model and tree structure on three different benchmarks, opening up opportunities for further speed up with SSM and hybrid model inference. Code can be find at: `https://github.com/wyc1997/stree`.

## 1 Introduction

Recursive sequence models, such as autoregressive (AR) Transformers, produce a single token with each forward pass. Speculative decoding is a technique to make this process more efficient by leveraging a smaller 'draft model' to generate multiple tokens, and a separate 'verifier model' to validate the proposed drafts [13]. The efficiency stems from exploiting the concurrency of parallel computer hardware to verify multiple tokens in a single model call, while ensuring that the accepted drafts are identical to those that would have been generated by repeated calls to the original model. AR Transformers leverage a sliding window of input tokens (context) as their 'state', which is ideal for speculative decoding since their context can be easily edited to remove unverified tokens.

On the other hand, State-Space Models (SSMs) maintain an explicit Markov state that is designed and trained to be a sufficient statistic of the past for the purpose of predicting one-step ahead [17]. They are not naturally suited for speculative decoding, since the past states are discarded and the verifier would need to backtrack each speculated state, hampering the efficiency of the whole process. Recently, hardware-aware efficient methods for speculative decoding in SSMs have been proposed [24, 25], but they do not leverage tree-based verification that drives the most efficient methods for

---

[*]Correspondence to `wuyangchao1997@g.ucla.edu`

39th Conference on Neural Information Processing Systems (NeurIPS 2025).

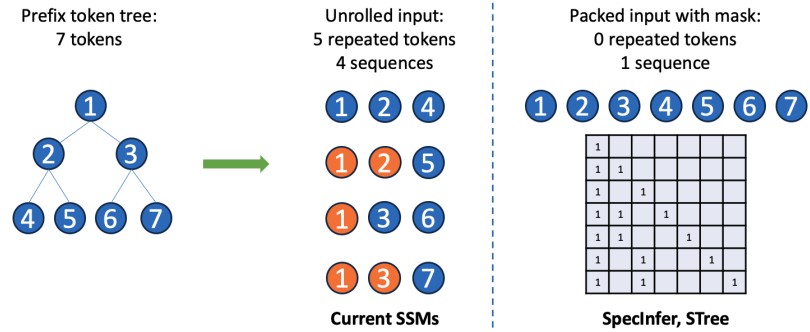

Figure 1: Methods to decode a prefix token tree with AR Transformers. A prefix token tree can be unrolled into multiple sequences (used by current SSMs) and computed as a batch or packed into one sequence using a mask to indicate the tree structure (first used by SpecInfer [19] and extended to SSMs here). The former leads to inefficiency due to repeatedly computed tokens (in orange).

AR Transformers. This paper proposes the first algorithm to do so, with a method that is applicable to both SSMs and hybrid architectures interleaving SSMs and Transformer layers.

To perform tree-based verification in AR Transformers [19, 14, 4, 2, 23, 21, 22], a prefix token tree is built with the draft model and verified with the target model. Existing works pack the token tree into one sequence and leverage a topology-aware mask [19] with self-attention to compute the output of the tree in one model call (Fig. 1 Right), thus avoiding repeated computation when naively unrolling the tree into individual sequences.

However, modern SSM realizations [6, 9] are designed to scan through all tokens in the past causally to obtain the state, lacking a mechanism to specify the tree structure. As a consequence, current SSMs can only use the unrolled input (Fig. 1 Left), leading to inefficient repeated token generation. Moreover, SSMs require one state per input sequence in the batch, leading to an explosion of the memory footprint when using the unrolled input. As the size of the tree grows, unrolling the tree quickly becomes infeasible. Therefore, we seek ways to leverage the characteristics of the hardware to perform multiple steps of state update following the tree structure through a single pass through the model.

We propose State-Space Speculative Tree (STree) decoding, a method to facilitate tree-based speculative decoding in SSMs through accumulating state transition matrices according to the tree structure. The tree is easily computed with minimal overhead relative to current SSM state update implementations. We describe a hardware-aware implementation that already in its simplest instantiation shows improvement over the baseline naive application of tree-based speculative decoding to SSMs.

Our contributions in this paper are to (i) propose what, to the best of our knowledge, is the first scalable method to leverage tree decoding in the speculative decoding for both SSMs and hybrid architectures; we also (ii) provide a simplified analysis of the trade-off between acceptance length and model runtime to help determine whether we should scale tree size or even use tree decoding. Finally, we (iii) empirically demonstrate that with a baseline drafting model and static tree structure, there are already improvements in generation speed, thus opening the door to further investigation of more advanced speculative decoding methods employed with transformers.

## 2   Related Work

**State-space models (SSMs)**   are sequence models that maintain a hidden state and use it to predict the next datum in the sequence given the previous ones [17]. Such models can be stacked, so the output of one is the input of another [10, 26], and their parameters can be input-dependent [12, 9, 27]. In order to scale model size, transition matrices are often restricted to be diagonal [6, 7], while to leverage efficient hardware-aware implementations, SSM layers are often interleaved with Transformers in Hybrid architectures [16, 8], also useful for initial distillation [24]. Our method is

based on scalable SSMs of the general form (1).

$$\begin{cases} x_{t+1} = A(u_t)x_t + B(u_t)u_t \\ y_t = C(u_t)x_t + D(u_t)u_t \end{cases} \tag{1}$$

**Speculative decoding**   denotes a set of techniques designed to parallelize sequential inference in large language models (LLMs) by utilizing a smaller 'draft model' to produce multiple candidate trajectories and a large 'verifier model' to test them for consistency with the original model [13]. Sub-networks can also be used for drafting, a form of 'self-speculation' [18], while speculation can be applied to latent features [15, 14] in addition to the output. Structural changes to the speculative processes include tree-based verification [20, 4], multi-head decoding [2], and beam sampling[21, 22]. The same methods can also be used for 'lossy verification' using judge decoding [1].

**Tree-based verification**   [19] has been shown effective in improving the acceptance length of the drafted sequence [15, 14, 4, 2, 23, 21, 22]. EAGLE [15] reported an increase in acceptance length by using a static tree structure, leading to an overall speed up ratio improvement. Sequoia [4] generates the optimal tree for the speculated tokens using a dynamic programming algorithm and achieved further improvement. The underlying mechanism that enables tree-based verification is Transformers' ability to specify which tokens to attend to with an arbitrary attention mask individually for each token. A prefix token tree can be packed into a sequence and a topology aware mask [19] can be used to inform the self attention block the tree structure.

**Speculative SSMs**   have been independently championed by [24, 25]. Since the state in SSMs is updated causally to summarize the past history, extending speculative decoding to SSMs require backtracking the state, which is non-trivial at scale. Both [24, 25] proposed hardware-aware efficient algorithms to backtrack the state while performing the forward pass. However, neither leveraged tree-based verification, since currently available algorithms for SSM updates require unrolling the tree into individual sequences, which leads to inefficiencies. These include repeated token computation and extra states to be maintained.

## 3   Method

**Formalization.**   Given a prefix token tree $T$ with tokens $\{t_1, \ldots, t_N\}$ as vertices and $t_1$ as the root node, we can pack the tokens into one sequence $S = \{t_1, \ldots, t_N\}$ and represent the topology of the tree using a special attention mask $L \in \{0, 1\}^{N \times N}$ ('tree' mask). Each vertex in $T$ has one unique path to the root node, and we denote these paths as $s_i = \{t_n | t_n$ is in the path from $t_1$ to $t_i\}$. Then the tree mask $L$ can be constructed by the indicator function:

$$L_{i,j} = \mathbb{1}_{s_i}\{t_j\}. \tag{2}$$

Given the pre-SSM input features $u = \{u_1, \ldots, u_N | u_i \in \mathbb{R}^{d_u}\}$ of the packed sequence $S$ and an initial state $x_0 \in \mathbb{R}^{d_x}$, our goal is to compute the output sequence $y = \{y_1, \ldots y_N | y_i \in \mathbb{R}^{d_u}\}$ without repeatedly computing any tokens or requiring extra SSM states. We present our approach below.

**Tree decoding with State-space models.**   The input feature $u$ is first mapped onto the parameters using a linear projection. Here we abuse the notation to let $X_t := X(u_t)$:

$$A_t = W_A u_t \in \mathbb{R}^{d_x \times d_x} \quad B_t = W_B u_t \in \mathbb{R}^{d_x \times d_u} \quad C_t = W_c u_t \in \mathbb{R}^{d_u \times d_x} \tag{3}$$

Ignoring the last term $D_t$ in Eqn.(1) for simplicity of notation, for each $y_t = C_t x_t$, we can expand it recursively into:

$$y_t = C_t(A_t^{L_{t,t}} A_{t-1}^{L_{t,t-1}} \ldots A_1^{L_{t,1}} x_0 + L_{t,1} A_t^{L_{t,t}} A_{t-1}^{L_{t,t-1}} \ldots A_2^{L_{t,2}} B_1 u_1 + \cdots + B_t u_t)$$

$$= C_t(A_t^{L_{t,t}} A_{t-1}^{L_{t,t-1}} \ldots A_1^{L_{t,1}} x_0 + \sum_{s=1}^{t} L_{t,s} (\prod_{j=s+1}^{t} A_j^{L_{t,j}}) B_s u_s) \tag{4}$$

As done in [9, 6, 7, 16], we enforce a diagonal structure on $A_i$ to reduce the products in Eqn.(4) to a sum of logarithms:

$$y_t = C_t(\exp\{\sum_{i=1}^{t} L_{t,i} \log(A_i)\} x_0 + \sum_{s=1}^{t} L_{t,s} \exp\{\sum_{j=s+1}^{t} L_{t,j} \log(A_j)\} B_s u_s)$$

$$= C_t \exp\{\sum_{i=1}^{t} L_{t,i} \log(A_i)\} x_0 + \sum_{s=1}^{t} L_{t,s} \exp\{\sum_{j=s+1}^{t} L_{t,j} \log(A_j)\} C_t B_s u_s \qquad (5)$$

Since $A_i$ is diagonal, we can represent the diagonal of $\log A_i$ as a vector and assemble a new matrix $A_{log} = [diag(\log A_1), \ldots, diag(\log A_N)]^T \in \mathbb{R}^{N \times d_x}$. We define:

$$A_{tree} := (L A_{log}) \quad (A_{tree})_t = \sum_{i=1}^{N} L_{t,i} \times diag(\log A_i) \in \mathbb{R}^{d_x} \qquad (6)$$

We note that $(A_{tree})_t$ is equivalent to $\sum_{i=1}^{t} L_{t,i} \log(A_i)$ in Eqn.(5), since $A_i$ is diagonal and we can always organize the tokens in the sequence such that $L_{t,i} = 0 \quad \forall i > t$. Furthermore, $\sum_{j=s+1}^{t} L_{t,j} \log(A_j)$ is equivalent to $(A_{tree})_t - (A_{tree})_s$. Then, Eqn.(5) becomes:

$$y_t = C_t(\exp\{(A_{tree})_t\} \circ x_0) + \sum_{s=1}^{t} L_{t,s} \exp\{(A_{tree})_t - (A_{tree})_s\} \circ (C_t B_s u_s)$$

where $\circ$ denotes element-wise multiplication. Then the entire sequence of outputs $y$ can be written as:

$$y = STree\_SSM(L, A, B, C)(x_0, u) = M_x x_0 + M_u u$$

where:

$$(M_x)_i = C_i \times diag(\exp\{(A_{tree})_i\})$$
$$(M_u)_{ij} = L_{ij} C_i B_j \times diag(\exp\{(A_{tree})_i - (A_{tree})_j\})$$

We note that our method is a generalization of the matrix transformation form of SSM in Mamba2 [6]. When $L$ is a lower triangular causal attention mask, our method is the same as [6] with a non-zero initial state. We introduced the quantity $A_{tree}$, which accumulates the state transition matrices $A$ according to the tree structure, enabling tree decoding with SSM. The computation for $A_{tree}$ is simple, and incorporating it into the original SSM imposes little overhead. We further note that our method is not limited to a tree structure, but can potentially be used with an arbitrary mask to specify the structure, allowing greater flexibility with SSMs.

**Implementation.** Based on the methods presented in previous paragraphs, we propose an algorithm in Alg. 1 to perform State-space speculative decoding with tree decoding. The algorithm centers on a tree scan kernel that computes the output of a packed prefix tree input and also caches the intermediate activation values (*i.e.*, $A_{i:j}, B_{i:j}, C_{i:j}$):

$$t'_{i:j}, Cache \leftarrow \text{TREESCAN}(L, t_{i:j}, x_i)$$

The kernel is hardware-aware as it avoids instantiating any intermediate results as well as the SSM states off from the fast GPU shared memory. During the verification process, we only generate the output $y_{i:j}$ for the input sequence, but not the state after the input. This is because, for speculative decoding, the state at the end of an input sequence is most likely incorrect unless all tokens in the input sequence are accepted. We use the activation replay method [25] to recompute the correct state before the first rejected tokens at the start of the next iteration.

## 4   Analysis

The wall time of speculative decoding is jointly determined by the runtime of the target model $t$ and the average acceptance length $\tau$. Specifically, we have

$$\text{Wall Time} \propto \frac{t}{\tau}$$

**Algorithm 1** Speculative decoding with Tree Scan for SSMs

---

1: **function** SPECULATIVEDECODINGWITHTREESCAN
2:     **Initialize** $L^*$ : mask to indicate last accepted token
3:     **Initialize** $Cache$ : activation cache to facilitate recomputation of state
4:     **Initialize** $x^*$ : the correct state
5:     **while** $should\_continue$ **do**
6:         $L_{i:j}, t_{i:j} \leftarrow \text{DRAFT}(t_{i-1})$           ▷ Draft a tree with last accepted token
7:         $x^* \leftarrow \text{ACTIVATIONREPLAY}(L^*, Cache)$   ▷ Recompute state up to the rejected tokens
8:         $t'_{i:j}, Cache \leftarrow \text{TREESCAN}(L, t_{i:j}, x^*)$   ▷ Getting output and cache from target model
9:         $L^*, t_{i:k} \leftarrow \text{FIRSTREJECTED}(t'_{i:j}, t_{i:j})$        ▷ Accept/Reject Drafted tokens
10:    **end while**
11: **end function**

---

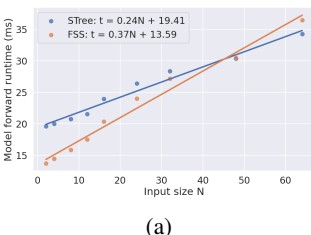 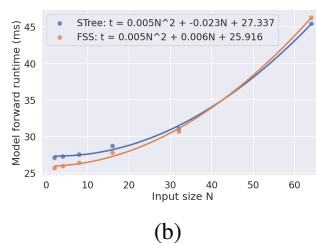 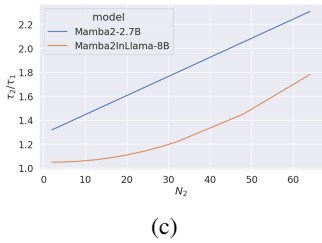

       (a)                   (b)                   (c)

Figure 2: **Left**: The runtime for a call to Mamba2-2.7B model vs. input size with STree and Fuse Selective Scan (FSS). A linear regression is performed to obtain the slope and intercept. **Middle**: The runtime for a call to MambaInLlama-8B model vs. input size with STree and FSS. A polynomial regression with degree 2 is used to obtain the parameters. **Right**: The ratio of acceptance length $\tau$ required to get wall-clock time improvement vs. input length $N_2$, with a fixed $N_1 = 5$

The success of speculative decoding relies on the fact that computing an input of length $N > 1$ doesn't increase $t$ too much. As SSMs are not as good as transformers in performing parallel computation due to their recurrent nature, the runtime of the target model calls sees a non-negligible increase even at small input length. Therefore, there exists a trade-off between the gain in acceptance length and the increase in runtime of target model calls as the size of the tree grows. Given that SSM compute and memory cost is linear in the input length, we assume a simple linear model between the model runtime and the input size, where $t = kN + C$, where $C$ is a constant contributed by the time for loading model weights and $kN$ is proportional to input size $N$, contributed by loading inputs and computation. When the models are large, the constant $C$ dominates the runtime, leading to a negligible effect on runtime by input length $N$ when $N$ is small.

Therefore, to gain wall-clock time improvement we require:

$$\frac{k_1 N_1 + C_1}{\tau_1} \geq \frac{k_2 N_2 + C_2}{\tau_2}$$

$$\frac{\tau_2}{\tau_1} \geq \frac{k_2 N_2 + C_2}{k_1 N_1 + C_1} \tag{7}$$

We note that when applying the above analysis to hybrid models, one key difference is that the transformer blocks in the hybrid models have quadratic space and time complexity in input length, and therefore would require a quadratic model $t = aN^2 + bN + c$ for modeling the runtime and input size, which leads to:

$$\frac{\tau_2}{\tau_1} \geq \frac{a_2 N_2^2 + b_2 N_2 + c_2}{a_1 N_1^2 + b_1 N_1 + c_1} \tag{8}$$

To determine whether we should use tree scan for speculative decoding, we want to evaluate whether the gain in acceptance length can outweigh the increase in runtime. For example, we measured the runtime for Mamba2-2.7B and MambaInLlama-8B model calls using both STree and Fused Selective Scan (FSS), which is an algorithm optimized for short sequence inference [25], with different input sizes in Fig. 2a and Fig. 2b. We performed linear and quadratic regression to obtain an approximation

Table 1: Latency of a Mamba2 forward pass using STree vs. autoregressive forward with selective scan. Note that for 7B, 13B, and 23B models, we initialize the model with random weight to measure the runtime since there is no pretrained model of those sizes.

|                | 2.3B          | 7B            | 13B           | 23B           |
|----------------|---------------|---------------|---------------|---------------|
| Autoregressive | 10.95         | 22.94         | 40.69         | 72.41         |
| STree          | 22.36 (2.04x) | 33.79 (1.47x) | 55.90 (1.37x) | 91.08 (1.26x) |
| Vanilla SD     | 15.05 (1.37x) | 26.46 (1.15x) | 45.95 (1.13x) | 76.74 (1.06x) |

of the parameters $(k, C, a, b, c)$. Assuming that we are comparing STree against vanilla state-space speculative decoding that uses FSS to compute forward pass where we draft 1 sequence with 5 tokens, we can let $N_1 = 5$ and plot the ratio of acceptance length $\tau_2/\tau_1$ required to achieve wall-clock time improvement for both models in Fig. 2c. We can see that for STree with Mamba2-2.7B, to gain a wall-clock time improvement at a tree size $N_2$ of 15 tokens, we need the drafted tree to achieve at least $1.5\times$ acceptance rate as compared to vanilla speculative decoding. On the other hand, STree with Mamba2InLlama-8B requires at least $1.1\times$ acceptance rate with the same tree size, which is much easier to achieve than Mamba2-2.7B. This means that the performance gain that can be achieved with tree decoding with Mamba2InLlama-8B is much more than that with Mamba2-2.7B. This could be due to Mamba2InLlama-8B being a larger model with a larger constant overhead for model call.

**Models of bigger size**   Theoretically, the effectiveness of speculative decoding depends on the runtime difference between the draft model and the target model as well as the acceptance rate of the drafted tokens. Everything else being equal, using a bigger target model leads to a bigger runtime difference. Therefore, the effectiveness of our method on bigger models should hold or improve. In Table 1, we provide the results of latency of a Mamba2 forward pass using STree vs. autoregressive forward with selective scan with same input tree configuration. Note that for 7B, 13B, and 23B models, we initialize the model with random weight to measure the runtime since there is no pretrained model of those sizes. We can see that as the model size grows, the relative overhead of using STree decreases from 2.04x to 1.26x, signaling that as model sizes increase, our method is likely going to be even faster, given that the average acceptance length stays the same. Meanwhile, the gap between the Vanilla SD and STree is closing as model size increases (from 48.9% with 2.3B to 18.9% with 23B), which is another signal that our method is scalable to larger models.

## 5   Experimental Results

In this section, we aim to demonstrate the efficiency of STree on speculative decoding. In Sec. 5.1, we compare STree against the unrolled input baseline to show that STree is more efficient at decoding a tree with SSMs. Then, in Sec. 5.2, we compare against speculative decoding without tree-based verification to show that STree is able to achieve speed improvement with baseline drafting models and tree construction methods. All experiments are run on an Nvidia RTX 3090 GPU.

### 5.1   Efficiency of STree against unrolled baseline

**Forward pass runtime.**   Since there is no efficient algorithm currently available for computing tree decoding with SSMs, we evaluate STree against the baseline method of unrolling the token tree into different sequences and computing the output of these sequences using the currently available Fused Selective Scan (FSS) [25], and Chunk Scan [6] kernels. We measure the runtime of a forward pass through a Mamba2-2.7B model. For the input token tree, we use full binary trees of 4/5/6 layers deep, which contain 15/31/63 tokens in the tree respectively. The results are shown in Fig. 3 Right.

STree performs on par with FSS at small tree size (4-layer), as FSS is optimized for short sequences. However, as the size of the tree increases, the forward pass with STree increases slightly from 27.6ms to 34.0ms, while FSS increases drastically from 27.3ms to 59.8ms. This is due to unrolling a prefix tree introduced repeated tokens and extra states as shown in Fig. 3 Left. This slows down the forward pass for both FSS as well as chunk scan. As STree is able to directly decode a packed tree sequence, it demonstrates good scalability as the input tree size grows because it avoids repeated computations.

| Tree depth | Methods | #. of States | Tokens Computed |
|---|---|---|---|
| 4 | Packed | 1 | 15 |
|   | Unrolled | 8 | 32 |
| 5 | Packed | 1 | 31 |
|   | Unrolled | 16 | 90 |
| 6 | Packed | 1 | 63 |
|   | Unrolled | 32 | 192 |

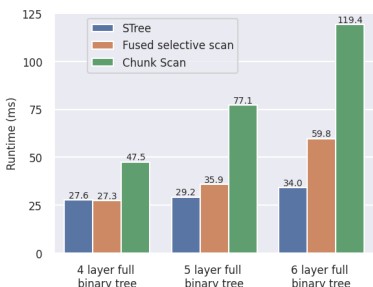

Figure 3: **Left**: Comparison of number of states required and number of tokens computed to get an output for a full binary tree in one forward pass. STree is able to decode a packed tree sequence, while other methods need to unroll the tree into multiple sequences. **Right**: Runtime in milliseconds (ms) for a forward pass for different full binary trees using different algorithms.

Table 2: Generation speed (tokens/second) and memory usage (GB) using STree on packed tree input and Fused Selective Scan (FSS) on unrolled tree input. M is the number of beams we keep at each step and N is the number of steps we beam-searched for. M=4/5 with N=16 for FSS ran Out Of Memory (OOM) on a 3090 GPU with 24GB of GPU memory.

|  |  | M=2 | | M=3 | | M=4 | | M=5 | |
|---|---|---|---|---|---|---|---|---|---|
|  | N | STree | FSS | STree | FSS | STree | FSS | STree | FSS |
| Speed | 4 | **102.69** $(1.05\times)$ | 97.44 | **101.68** $(1.12\times)$ | 90.64 | **94.61** $(1.10\times)$ | 85.33 | **91.80** $(1.16\times)$ | 78.94 |
|  | 8 | **107.28** $(1.09\times)$ | 97.53 | **104.99** $(1.21\times)$ | 86.68 | **100.67** $(1.31\times)$ | 76.72 | **98.75** $(1.44\times)$ | 68.44 |
|  | 16 | **105.62** $(1.25\times)$ | 84.26 | **101.97** $(1.49\times)$ | 68.14 | **94.47** | OOM | **88.41** | OOM |
| Memory | 4 | **7.67** $(0.91\times)$ | 8.47 | **7.68** $(0.85\times)$ | 8.97 | **7.73** $(0.83\times)$ | 9.36 | **7.75** $(0.78\times)$ | 9.85 |
|  | 8 | **7.79** $(0.86\times)$ | 9.05 | **7.85** $(0.80\times)$ | 9.76 | **7.88** $(0.72\times)$ | 10.89 | **7.91** $(0.64\times)$ | 12.32 |
|  | 16 | **8.02** $(0.75\times)$ | 10.69 | **8.12** $(0.57\times)$ | 14.02 | **8.17** | OOM | **8.28** | OOM |

Meanwhile, forward pass with chunk scan is consistently slower than both fused selective scan and STree, as it is optimized for parallel training for long sequences and pays a big overhead at a short sequence length [25, 24]. Hence, it should not be considered for use in speculative decoding.

**Generation speed against unrolled baseline.** Taking one step further, we measure the generation speed of STree with packed tree input against FSS with unrolled tree input. We used a Mamba2-2.7B model as the target model and a Mamba2-130M model as the drafting model. We generate the token tree using beam search [11, 23], where we perform beam search with the drafting model and keep all the tokens generated at each step of beam search, even if the beam is later discarded. This results in an N-layer tree with M tokens at each layer, where N is the number of beam search steps and M is the number of beams. We verify the target model with greedy search, where at each step, the token with the maximum conditional likelihood from the target model is compared to the corresponding child nodes in the token tree to see if we should accept the draft. The acceptance length for the two methods is the same, as the same drafted tree is used for verification. We perform this experiment on the MT_Bench [28] benchmarks for generating 100 tokens. The results are presented in Tab. 2.

We can see that STree with packed input is both faster and more memory efficient than FSS with unrolled input across all tree sizes. We also notice that as the size of the tree grows, the advantage with STree becomes bigger ($1.05\times$ to $1.49\times$ for speed and $0.91\times$ to $0.57\times$ for memory), which coincides with our previous finding that STree forward pass is more scalable to larger trees. Meanwhile, we note that unrolling the tree also adds overhead to the execution. Memory-wise, the increase in memory for STree is almost negligible when tree size increases, mainly due to the increase in input size. On the contrary, the repeated tokens and states required by FSS contribute to a large increase in GPU memory, making a large token tree ($N = 16, M = 4/5$) infeasible to compute.

Table 3: Generation speed (tokens/sec.) and average number of tokens accepted $\tau$ for Vanilla speculative decoding (SD) and STreewith MambaInLlama-8B model with 50% mix of transformers.

| Methods | MT-Bench | | HumanEval | | GSM-8K | |
|---|---|---|---|---|---|---|
| | Speed | $\tau$ | Speed | $\tau$ | Speed | $\tau$ |
| Temperature=0 | | | | | | |
| Autoregressive | 39.98 | 1 | 39.98 | 1 | 40.47 | 1 |
| Vanilla SD | 67.68 (1.69×) | 2.04 | 77.18 (1.93×) | 2.32 | 78.38 (1.93×) | 2.34 |
| STree | **69.84** (1.74×) | 2.47 | **78.35** (1.95×) | 2.79 | **80.03** (1.98×) | 2.83 |
| Temperature=1 | | | | | | |
| Autoregressive | 39.72 | 1 | 40.04 | 1 | 40.16 | 1 |
| Vanilla SD | 49.24 (1.23×) | 1.55 | 52.88 (1.32×) | 1.65 | 51.80 (1.28×) | 1.61 |
| STree | **54.08** (1.36×) | 2.03 | **60.34** (1.50×) | 2.26 | **58.25** (1.45×) | 2.16 |

## 5.2 End-to-end generation with hybrid models

**Generation with MambaInLlama models.** Having verified the effectiveness of STree in performing tree decoding, we then demonstrate the potential of STree in further boosting the speed of generation for SSMs and hybrid models. We choose a Mamba2InLlama-8B[2] [24] hybrid model, with 50% mix of transformers and SSMs, as the target model. As there is a lack of smaller models in the same family, we distill a 2-layer SSM from the target model using data that is used to finetune the target model. We show the results of using a checkpoint from 48000 steps into the distillation. Results from different checkpoints are shown in the ablation study below.

For the vanilla speculative decoding baseline, we use the 2-layer model as the drafting model to draft 1 sequence of 4 tokens every step (target input size $1 \times 5$) and verify with the target model output using speculative sampling algorithm [**?** ]. For STree, we use the draft model to draft a static tree structure shown in Fig. 4a, and use multi-step speculative sampling (MSS sampling) [19] to verify with the target model output. Both methods use activation replay to backtrack the state. We evaluate the speed of generating 1024 tokens on three different benchmarks: MT_Bench [28], HumanEval [3], and GSM8K [5], with two different temperatures: 0 (greedy) and 1. The results are shown in Tab. 3

We can see that across all three benchmarks used, STree is able to outperform vanilla speculative decoding. The advantage becomes more obvious when the temperature of the generation increases and the acceptance length drops as a result. We note that this performance improvement is achieved with a baseline tree generation strategy (a static tree) and a draft model still early in its distillation process. With a more advanced tree generation strategy and draft model, the acceptance length with STree will likely improve, which will translate into more generation speed improvement. We include the results with H100 GPUs and a beam search tree in the appendix due to space constraint.

**Ablation studies**

**Effect of temperature** We study the effect of sampling temperature on the acceptance rate and speed-up. We use the same setup as the end-to-end generation experiment with MambaInLlama model and vary the temperature used with sampling. From Fig. 4b and Fig. 4c, we can see that average acceptance length drops as temperature goes up for both STree and vanilla speculative decoding, leading to a decrease in generation speed. We notice that the absolute difference between the acceptance length of STree and vanilla speculative decoding is relatively stable ($\sim 0.4$), which means that as the acceptance length drops for both STree and vanilla speculative decoding, the improvement in generation speed becomes bigger, re-affirming our results in the previous sections.

**Effect of sampling algorithm** Besides multi-step speculative sampling, we test our method with naive sampling used in [19, 23]. Specifically, we use the top-k tokens from the drafting distribution to build our token tree, and perform sampling with the target model. If at any token, the sampled token from the target model falls within the top-k tokens drafted, we accept that token and continue to verify

---

[2]Checkpoint at https://huggingface.co/JunxiongWang/Llama3.1-Mamba2-8B-distill

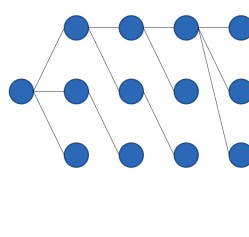 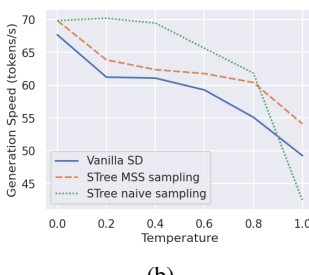 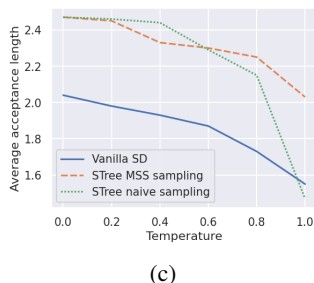

| (a) | (b) | (c) |

Figure 4: **Left**: Structure of static tree used to generate a prefix token tree with the drafting model. We draft 4 steps for each iteration and 3 tokens for each layer, resulting in 13 tokens in every input sequence. **Middle**: Generation speed of STree and Vanilla Specultaive Decoding (SD) under different temperature. **Right**: Average acceptance length of STree and Vanilla Speculative Decoding (SD) under different temperature.

Table 4: Inference speed (tokens/sec.) and average number of tokens accepted $\tau$ for STree with MambaInLlama-8B models with different static tree configurations (sampling temperature = 1).

| Static tree configuration | A | B | C | D | E |
|---|---|---|---|---|---|
| Max tree width | 3 | 16 | 6 | 4 | 3 |
| Tree depth | 4 | 3 | 3 | 4 | 5 |
| Number of tokens | 13 | 29 | 16 | 13 | 16 |
| Speed | 54.08 | 46.90 | 51.19 | 54.39 | 56.79 |
| $\tau$ | 2.03 | 2.15 | 2.01 | 2.07 | 2.09 |

the next token. We refer the readers to [23] for more details. The results are shown together in Fig. 4b and Fig. 4c. We can see that at low temperatures, naive sampling has a relatively high acceptance length and fast run time. As temperature increases, the acceptance length drops significantly, because the sampled token by the target model is less likely to come from the top-k tokens with the drafted model. This leads to a drop in overall generation speed as well. We note that this drop in generation speed is due to the characteristic of the sampling algorithm, but not our framework. With a high acceptance length at lower temperatures, we can still achieve speed improvements.

**Effect of static tree structure**   We study the effect of different static tree structures on the acceptance rate and speed-up. We note that producing a wider and deeper tree also slows the drafting model and contributes to the overall runtime. The static tree configurations are shown in Fig. 5 in the Appendix. The results are presented in Tab. 4. To determine whether to use a deeper/wider tree, we need to consider not only the effect on the acceptance rate, but also the runtime of the target and drafting models. Increasing the width of the tree drastically does improve the acceptance rate (Column B vs. C), but it also slows down the models and therefore overall speed. When we increase the tree depth (Column A vs. E), we can see that the improvement in $\tau$ outweighs the runtime cost, and therefore we see an improvement. Comparing Column A and D, where the number of tokens is the same but with different connectivity, improvement can be seen in both speed and $\tau$ when a better tree structure is used. This signifies the importance of having a good tree structure in using STree. This result also agrees with our analysis on the trade-off between runtime and acceptance rate. It provides the promise that when we use a better-trained drafting model and better drafting algorithms, which would give us a better tree and token candidates, the generation speed with STree will improve.

**Effect of percentage of transformer blocks in target model**   Hybrid models are a mix of transformer blocks and SSM blocks. As Transformers have better scaling in parallel compute, we hypothesize that more transformer blocks in the model will lead to a larger improvement in generation speed by STree. We perform an ablation study using Mamba2-Llama3 models [24] with 50%, 25%, and 0% of transformer blocks, using the same drafting model we distilled. The results are shown in Tab. 5. With autoregressive generation, models with more SSM blocks are faster, which agrees with our expectation. For generation with STree, we are still able to outperform vanilla speculative decoding using models with fewer transformer blocks. If we eliminate the effect of different acceptance

Table 5: Inference speed (tokens/sec.) and average number of tokens accepted $\tau$ for Vanilla speculative decoding (SD) and STreewith Mamba2-Llama3 models with sampling temperature of 1. The percentage number indicated with the model name is the percentage of transformer blocks in the model. $\frac{\text{Speed-up}}{\tau}$ is obtained by dividing ratio of speed-up against auto-regressive speed in the bracket by $\tau$

| | Mamba2-Llama3 (50%) | | | Mamba2-Llama3 (25%) | | | Mamba2-Llama3 (0%) | | |
|---|---|---|---|---|---|---|---|---|---|
| Methods | Speed | $\tau$ | $\frac{\text{Speed-up}}{\tau}$ | Speed | $\tau$ | $\frac{\text{Speed-up}}{\tau}$ | Speed | $\tau$ | $\frac{\text{Speed-up}}{\tau}$ |
| Autoregressive | 39.71 | 1 | - | 41.65 | 1 | - | 43.79 | 1 | - |
| Vanilla SD | 46.56 (1.17×) | 1.47 | 0.80 | 47.69 (1.14×) | 1.45 | 0.78 | 53.72 (1.22×) | 1.57 | 0.77 |
| STree | **49.04** (1.23×) | 1.84 | 0.66 | **48.08** (1.15×) | 1.77 | 0.64 | **55.32** (1.26×) | 2.00 | 0.63 |

Table 6: End-to-end generation speed (in Tokens/seconds) and acceptance length for Vanilla Speculative decoding and STree using drafting model trained for different steps.

| | 12000 step | 48000 step | 264000 step |
|---|---|---|---|
| Speed (Vanilla SD) | **44.52** (1.21x) | 53.11 (1.24x) | 56.49 (1.42x) |
| $\tau$ (Vanilla SD) | 1.34 | 1.56 | 1.68 |
| Speed (STree) | 42.96 (1.19x) | **58.65** (1.36x) | **64.30** (1.62x) |
| $\tau$ (STree) | 1.51 | 2.02 | 2.23 |

lengths in the $\frac{\text{Speed-up}}{\tau}$ column, we can see that models with fewer transformers see less speed-up per acceptance length (0.66 to 0.63), agreeing with our hypothesis.

**Effect of different drafting model checkpoints**    In Table 6, we provide an ablation study on our drafting model distilled to different steps. We can see that as the distillation goes on, the acceptance length for the draft model gets longer. As the acceptance length increases, the end-to-end generation speed of STree improves from being 1% slower than Vanilla SD at step 12000 to 9.67% faster at step 48000 and a further 14.8% speedup at step 264000. This is because the distribution of the drafting model is more aligned with the target model, suggesting that with better drafting model and technique that gives a better acceptance length, the end-to-end runtime of STree will become better.

## 6   Discussion

Our work proposes STree, an efficient algorithm to unlock the potential of speculative tree decoding for SSMs and hybrid models. We close by discussing the limitations of STree. As presented in our analysis, STree has an overhead at a short input length as compared to previous methods, and will not universally improve generation speed when applied without careful consideration. Reducing this overhead through further algorithmic innovation could lead to a better trade-off. Meanwhile, we only demonstrate improvement on 8B models, which are still considered small LLMs. scaling up our method to bigger models and more advanced speculative decoding methods may bring more benefits, including energy savings for LLM inference. We leave that for the exploration of future work.

**Acknowledgements.** This work was supported by ONR award #N000142212252 and NSF 2112562 Athena AI Institute.

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

## A  Static Tree configurations

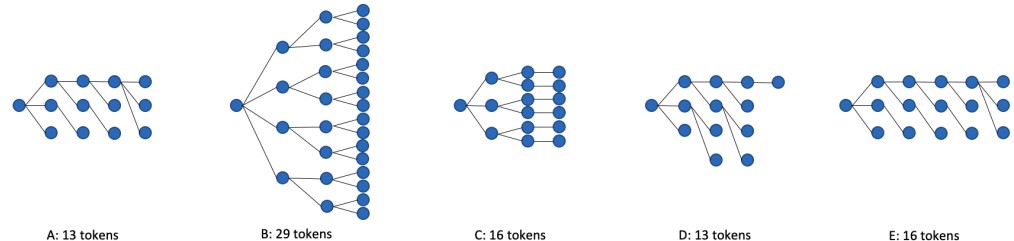

Figure 5: Static tree structure that we used in our ablation study for effect of differnt tree structure.

Fig. 5 show the configurations of the tree we tried in our ablation study. We attempted different breadth (B vs. C), different depth (A vs. E). same number of tokens but different connectivity (A vs. D). The results are show in the

## B  Experiments on H100 GPU

We further extend our method to H100 GPUs to demonstrate the our method is still applicable with better hardware. We apply our kernels as it is on H100 GPUs and compare it to vanilla speculative decoding.

Table 7: Generation speed (tokens/sec.) and average number of tokens accepted $\tau$ for Vanilla speculative decoding (SD) and STree with MambaInLlama-8B model with 50% mix of transformers on H100 GPU.

| | MT-Bench | | HumanEval | | GSM-8K | |
|---|---|---|---|---|---|---|
| Methods | Speed | $\tau$ | Speed | $\tau$ | Speed | $\tau$ |
| | | | Temperature=0 | | | |
| Autoregressive | 78.09 | 1 | 78.44 | 1 | 79.33 | 1 |
| Vanilla SD | **113.60** (1.45×) | 2.03 | **129.76** (1.65×) | 2.33 | **131.62** (1.66×) | 2.33 |
| STree | 113.13 (1.45×) | 2.45 | 127.15 (1.62×) | 2.77 | 131.13 (1.65×) | 2.81 |
| | | | Temperature=1 | | | |
| Autoregressive | 76.66 | 1 | 77.67 | 1 | 78.35 | 1 |
| Vanilla SD | 80.76 (1.05×) | 1.56 | 85.63 (1.10×) | 1.64 | 87.04 (1.11×) | 1.63 |
| STree | **85.18** (1.11×) | 2.04 | **92.74** (1.19×) | 2.22 | **92.86** (1.19×) | 2.16 |

From the above tables, we can see that the extent of improvement from applying both Vanilla SD and STree is smaller as compared to autoregressive decoding. This is because the memory bandwidth of H100 GPU is bigger than that of RTX3090, leading to a smaller bottleneck in memory transfer. With greedy generation, STree achieves a slightly slower speed as compared to Vanilla SD. When temperature=1, STree still holds an advantage. This is due to the relative increase in acceptance length is smaller when we are doing a greedy generation and the relative overhead between the STree and Vanilla Speculative decoding increases due to the faster GPU. We believe that as the model size increases and the memory bandwidth bottleneck due to the model weight transfer becomes more significant, we will still be able to show improvement. We also note that we present a general algorithm that is not GPU specific and is not optimized for H100 GPU.

## C  End-to-end generation result using beam search tree

We note that STree can work with any tree generation strategy and optimizing the tree structure is orthogonal to our work. Here, we further show that the end-to-end generation results using a beam search tree for drafting, which is a basic dynamic tree structure. The expansion of the tree depends on the joint probability and will vary from sample to sample. We use greedy decoding with M = 3 and N

Table 8: End-to-end generation speed and acceptance length ($\tau$) for different method of drafting.

| | Autoregressive | Vanilla SD | STree w. static tree | STree w. beam search tree |
|---|---|---|---|---|
| Speed | 39.98 | 67.68 (1.69x) | 69.84 (1.75x) | 71.60 (1.79x) |
| $\tau$ | 1 | 2.04 | 2.47 | 2.60 |

= 4 (M, N explained in table 1). The results are shown in Table 8. We can see that with a dynamic tree generation strategy, we are able to get a better acceptance length and thus a faster end-to-end generation speed. We believe that more advance tree construction technique would bring even more benefits.

