# OpenReview forum: "STree: Speculative Tree Decoding for Hybrid State Space Models"
_NeurIPS.cc/2025/Conference — NeurIPS 2025 poster_

### Official Review · Reviewer_7vpj · 2025-06-29

**Clarity:** 2
**Significance:** 2
**Originality:** 1
**Rating:** 4
**Confidence:** 4

**Summary:**

This paper extends the tree-based speculative decoding from transformer-based auto-regressive models to the mamba models (SSMs) or hybird architectures.
Authors build their method, STree, based on accumulated state transition matrices.
They design a hardware-aware TreeScan kernel for the naive application of AR Transformer tree-based speculative decoding methods to SSMs.
The proposed method can not only accelerate the inference of SSMs, but also reduce the memory usage during inference.
This work outperforms vanilla speculative decoding with SSMs on three different benchmarks.

**Questions:**

1. Is there any design to make the configuration of the tree in dynamic ways?

2. As compared to the Vanilla SD method, the work does not achieve obvious acceleration improvements, is there any latency and memory results for other scales like 370M, 790M, 1.4B, or other new mamba models.

3. In table 2, can you provide more results with T=2, 3, 4...

4. In table 2, can you provide more results with more benchmarks? like MMLU, HellaSwag, SWAG, BBH, BoolQ

5. Latency comparison on other powerful GPUs? like A100, H100

6. Ablation study for different draft models?

**Ethical Concerns:**

["NO or VERY MINOR ethics concerns only"]

**Final Justification:**

Although I think the proposed method designed for SSMs is not novel, as there is similar method designed for LLMs.

The authors' rebuttal soloved my questions with several experiments.

I think the paper could be borderline accepted.

**Limitations:**

1. The novelty of the work is limited.

2. The tree configuration is static, which limits the model performance

3. This work only tests results with RTX 3090, while the results with powerful GPUs like A100 and H100 remains unknown.

**Quality:**

2

**Strengths And Weaknesses:**

Strengths:

1. This paper extends the tree-based speculative decoding to mamba models

2. The special kernel TreeScan kernel is a hardware-aware design, which minimizes the memory reduction.

3. Compared to Fused Selective Scan (FSS), STree achieves faster inference and memory reduction.


Weakness:

1. Limited model scales, this paper only reported models: Mamba2-2.7B and MambaInLlama-8B.

2. Static tree configurations: the manually setting is required for the tree configuration.

3. The acceleration for the model inference compared to the Vanilla SD is not obvious.

---

> ### Author Rebuttal · Authors · 2025-07-30
>
> We thanks the reviewer for their reviews. We address the concerns and questions raised below:
>
> **W1**: See Q2.
>
> **W2**: See Q1.
>
> **W3**: See Q2.
>
> **Q1. Dynamic tree**: We note that optimizing the tree structure is orthogonal to our work. Our work focuses on enabling efficient computation of token tree with SSMs. With a baseline static tree, we are already able to obtain runtime improvement, as shown in Table 2 in paper. We also note that the beam search tree we used in Table 1 is not static, as the expansion of the tree depends on the joint probability and will vary from sample to sample. Here, we additionally show the end-to-end result for drafting with a beam search tree. We use greedy decoding with the M = 3 and N = 4 (M, N explained in table 1). The results are as follow:
>
>  |    | Autoregressive | Vanilla SD | STree w. static tree | STree w. beam search tree|
>  |:---|:-------|:--------|:-------|:--------|
>  | Speed |  39.98 | 67.68 (1.69x) | 69.84 (1.75x) | 71.60 (1.79x) |
>  | $\tau$ |   1 | 2.04 | 2.47 | 2.60 |
>
> We can see that with a beam search tree, we are able to get a better acceptance length and thus a faster end-to-end generation speed. We believe that more advance tree construction technique would bring even more benefits.
>
> **Q2: improvement not obvious**: At temperature=0, our STree is always better than Vanilla SD, though by a smaller margin. However, at temperature=1, our method achieves an average 12\% gain, which is an obvious difference. Averaging over all 6 experiment, there is still a 7\% gain, which is non-trivial. Meanwhile, we show the runtime for forward passes for bigger Mamba2 models. STree and Vanilla SD both pay an overhead over autoregressive runs to compute multiple tokens. As model size increase, the ratio between STree overhead and Vanilla SD overhead decreases from 1.48 to 1.19. This means that when we have a bigger model，it is much easier to achieve an end-to-end performance gain. Given that we already achieved end-to-end performance gain in 8B models as show in Table 2 in the paper, there is reason to believe that performance with bigger models will further improve with similar acceptance rate.
>
>  |    | 2.7B | 7B | 13B | 23B |
>  |:---|:-------|:--------|:-------|:--------|
>  | Autoregressive |  10.95 |  22.94 | 40.69 | 72.41 |
>  | STree |   22.36 (2.04x) | 33.79 (1.47x) | 55.90 (1.37x) | 91.08 (1.26x) |
>  | Vanilla SD |  15.05 (1.37x) | 26.46 (1.15x) | 45.95 (1.13x) | 76.74 (1.06x) |
>
> **Q3: Generation Temperature=2,3,4**: In practice, no previous works in speculative decoding has used generation with temperature beyond 1. Using too high a temperature can lead to high perplexity in generated output and therefore hinders downstream tasks. We have the ablation study for temperature between 0 and 1 in figure 4b and 4c in the paper. Here, we additionally provide experiment with even higher temperature=1.1 and 1.2. We can see that the acceptance length and hence generation speed continues to decrease with such high temperature.
>
> | MT_Bench | | |
>  |:---|:---:|:---:|
>  |                 | Vanilla SD | STree |
>  | Speed (T=1.1)  |49.90  |   55.25|
>  | $\tau$ (T=1.1)  | 1.59 | 2.07|
>  | Speed (T=1.2)  | 48.32 | 52.04|
>  | $\tau$ (T=1.2)  | 1.51 | 1.94|
>
> **Q4: More benchmarks**: The three benchmarks we provide in Table 2 (MT\_Bench, GSM8K, Human\_Eval) is extensively used by previous works to benchmark speculative decoding performance with text completion as the task. We note that MMLU, HellaSwag, SWAG, BBH, BoolQ are all datasets on common sense reasoning and Natural Language inference. Some of them feature multiple choice questions, requiring only a single word as answer, which can lead to premature termination in generation. This could lead to abnormal token distribution if we continue to generate beyond the eos token, affecting the acceptance length of both the draft and target model. This is probably why these datasets are often not used in speculative decoding works. Nonetheless, we additionally provide BoolQ's reults here. We note that as we employ a lossless verification strategy with temperature=0, the results we generated will be identical to the result of the target model and we only focus on the runtime. We use the question in BoolQ as a prompt and let it generate for 1024 tokens to measure the runtime.
>
>  |  BoolQ | |  |
>  |:---|:---:|:---:|
>  |                 | Speed | $\tau$ |
>  | Autoregressive  |39.98  |   1 |
>  | Vanilla SD      | 70.52 | 1.91|
>  | STree           | 71.35 | 2.29|
>
> **Q5: Larger GPUs**: While we do not have access to H100/A100 GPUs, we provide results obtained using A6000 GPU, which is the best we have. The results are presented below. We note that inference for SSM and hybrid models are likely still memory bound and computation speed is not the bottleneck. Therefore, faster computation units from better GPUs should not invalidate our improvements.
>
>  | | MT_Bench |  | HumanEval|  | GSM8K | |
>  |:---|:---|:---|:---|:---|:---|:---|
>  |                 | Speed | $\tau$ | Speed|   $\tau$ | Speed | $\tau$ |
>  |                 |       | T=0 |       |      |       |      |
>  | Autoregressive  |37.78  |   1 | 37.96 |   1  |       |  1   |
>  | Vanilla SD      | 64.24 (1.70x) | 2.06| 73.90 (1.95x) | 2.34 | 74.10 	(1.94x) | 2.33 |
>  | STree           | 65.71 (1.74x) | 2.42 | 74.81 (1.97x) | 2.77 | 76.47 (2.01x) | 2.81 |
>  |                 |       | T=1 |       |      |       |      |
>  | Autoregressive  |37.45  |   1 | 37.49 |   1  |       |  1   |
>  | Vanilla SD      | 47.09 (1.26x) | 1.57 | 49.71 (1.33x) | 1.66 | 49.30 | 1.63 (1.30x)|
>  | STree           | 51.80 (1.38x) | 2.04 | 56.57 (1.51x)	| 2.23 | 55.23 | 2.16 (1.46x)|
>
> **Q5: Draft model Ablation**: For MambaInLlama-8B model, there are no currently viable open-source drafting models that uses the same tokenizer. Therefore we distilled our own drafting model. If the reviewer would like to suggest any suitable model for drafting, we are happy do experiments. Here, we additionally provide an ablation study on our drafting model distilled to different steps.
>
>  |  | 12000 step | 48000 step | 264000 step |
>  |:---|:---|:---|:---|
>  | Speed (Vanilla SD) |   44.52 (1.21x)| 53.11 (1.24x) | 56.49 (1.42x) |
>  | $\tau$ (Vanilla SD)|   1.34           | 1.56          |  1.68       |
>  | Speed (STree) |   42.96 (1.19x)| 58.65 (1.36x) | 64.30 (1.62x) |
>  | $\tau$ (STree)|   1.51           | 2.02         |  2.23        |
>
> We can see that as the distillation goes on, the quality of the drafting model gets better, which is reflected by the increase in acceptance length. The result also suggest that with better drafting model and technique, the end-to-end runtime of STree will become better.
>
> **L1: Novelty**: To the best of our knowledge, we are the first to apply speculative tree decoding to SSMs and hybrid models, which makes our work novel. While tree based speculative decoding exists for transformer based models, it heavily relies on the transformer's property that computing a tree of tokens doesn't incur much overhead with the use of attention masks. Without this property, it is much slower to compute a tree of tokens in SSMs due to their recurrent state update. Our work addresses this critical challenge by improving the efficiency of computing a tree of tokens with SSMs, which is not done in previous works.
>
> **L2**: See Q1
>
> **L3**: See Q4

---

> > ### Comment · Reviewer_7vpj · 2025-08-05
> > **Response to authors' rebuttal**
> >
> > Thank you for the authors' rebuttal, and I think the rebuttal has solved my questions.

---

> > > ### Author Response · Authors · 2025-08-05
> > >
> > > Per the request of the reviewer, we purchased H100 computing resources and obtain some results using our current implementation as it is on h100. The setting is the same as table 2 mentioned in the paper and the A6000 results provided in the rebuttal. We can see that without any further optimization for h100 GPUs, we are already seeing a end-to-end speedup improvement for STree against Vanilla SD. This shows the potential of STree on better hardware. We will include more result with H100 GPUs in the future version of our paper.
> > >
> > >  | | MT_Bench |  | HumanEval|  | GSM8K | |
> > >  |:---|:---|:---|:---|:---|:---|:---|
> > >  |                 | Speed | $\tau$ | Speed|   $\tau$ | Speed | $\tau$ |
> > >  |                 |       | T=1 |       |      |       |      |
> > >  | Autoregressive  |76.66         |   1 | 77.67        |   1  |   78.35    |  1   |
> > >  | Vanilla SD      | 80.76 (1.05x)| 1.56| 85.63 (1.10x)| 1.64 | 87.04 (1.11x)| 1.63 |
> > >  | STree           | 85.18 (1.11x)| 2.04| 92.74 (1.19x)| 2.22 | 92.86 (1.19x)| 2.16 |

---

### Official Review · Reviewer_bAPx · 2025-06-30

**Clarity:** 3
**Significance:** 3
**Originality:** 2
**Rating:** 4
**Confidence:** 4

**Summary:**

This paper introduces State-Space Speculative Tree Decoding (STree), to perform tree-based speculative decoding in state-space models (SSMs) and hybrid SSM/Transformer architectures. By accumulating state-transition matrices according to a draft-generated prefix token tree, STree packs the entire tree into a single SSM forward pass (rather than unrolling it into many sequences), dramatically reducing redundant computation and memory footprint. the contribution is,

1. a concise derivation of the tree-accumulation mechanism,

2. a hardware-aware TreeScan kernel to implement it efficiently,

3. an analysis quantifying the trade-off between tree size and acceptance length, and

4. extensive experiments on pure SSMs and hybrid MambaInLlama models showing up to ~1.5× speed-ups and major memory savings over unrolled baselines.

**Questions:**

N/A

**Ethical Concerns:**

["NO or VERY MINOR ethics concerns only"]

**Limitations:**

See weaknesses

**Quality:**

3

**Strengths And Weaknesses:**

Strengths

S1. They extend tree-based speculative decoding—previously exclusive to Transformers—to SSMs and hybrid models, leveraging the diagonal structure of SSM transition matrices, they provide a step-by-step derivation of the tree accumulation and how it generalizes existing matrix‐SSM formulations.

S2. The evaluation is intensive. Forward‐pass benchmarks (Fig. 3): STree scales gracefully from 15→63 tokens (27.6 ms→34.0 ms) vs. a doubling for unrolled FSS (27.3 ms→59.8 ms). End-to-end generation (Table 2): On MT-Bench, HumanEval, GSM8K, STree delivers up to ~1.98× speed-up vs. autoregressive and ~1.36×–1.50× vs. vanilla speculative decoding, especially at higher temperatures.

Weaknesses

W1. Results report mean speeds and acceptance lengths but lack error bars or multiple runs.

W2. The experiments were only run on an NVIDIA RTX 3090 GPU, and it is unclear how they perform on other hardware—such as the H100 using fast GEMM optimizations. It should report the performance on H100 as well.

---

> ### Author Rebuttal · Authors · 2025-07-28
>
> We thank the reviewer for recognizing the presentation of method and evaluation. We address the concerns raised by the reviewer below:
>
> **W1: Multiple Runs**: Thank you for you suggestion. Results in table 1 and table 2 (temperature=0) is performed with greedy decoding so re-running the experiment for multiple runs will yield the same results modulo hardware fluctuations. For Table 2 (temperature=1), we additionally provide the average and standard deviation for 4 runs here. We will include that in the updated version of the paper.
>
>  | | MT_Bench |  | HumanEval|  | GSM8K | |
>  |:---|:---|:---|:---|:---|:---|:---|
>  |                 | Speed | $\tau$ | Speed|   $\tau$ | Speed | $\tau$ |
>  | Autoregressive  |39.72         |   1 | 39.83        |   1  |   40.16    |  1   |
>  | Vanilla SD      | 51.88 $\pm$ 1.92| 1.56 $\pm$ 0.01| 54.66 $\pm$ 1.24| 1.65 $\pm$ 0.00|55.02 $\pm$ 2.15 | 1.61 $\pm$ 0.0|
>  | STree           | 57.15 $\pm$ 2.16| 2.02 $\pm$ 0.01| 62.68 $\pm$ 1.68| 2.26 $\pm$ 0.02 | 61.69 $\pm$ 2.30 | 2.16 $\pm$ 0.0|
>
> **W2: Better GPU**: While we do not have access to H100/A100 GPUs, we provide results obtained using A6000 GPU, which is the best we have. The results are presented below. We note that inference for SSM and hybrid models are still memory bound and computation speed is not the bottleneck. Therefore, faster computation units from better GPUs will not invalidate our improvements.
>
>  | | MT_Bench |  | HumanEval|  | GSM8K | |
>  |:---|:---|:---|:---|:---|:---|:---|
>  |                 | Speed | $\tau$ | Speed|   $\tau$ | Speed | $\tau$ |
>  |                 |       | T=0 |       |      |       |      |
>  | Autoregressive  |37.78  |   1 | 37.96 |   1  |       |  1   |
>  | Vanilla SD      | 64.24 (1.70x) | 2.06| 73.90 (1.95x) | 2.34 | 74.10 	(1.94x) | 2.33 |
>  | STree           | 65.71 (1.74x) | 2.42 | 74.81 (1.97x) | 2.77 | 76.47 (2.01x) | 2.81 |
>  |                 |       | T=1 |       |      |       |      |
>  | Autoregressive  |37.45  |   1 | 37.49 |   1  |       |  1   |
>  | Vanilla SD      | 47.09 (1.26x) | 1.57 | 49.71 (1.33x) | 1.66 | 49.30 | 1.63 (1.30x)|
>  | STree           | 51.80 (1.38x) | 2.04 | 56.57 (1.51x)	| 2.23 | 55.23 | 2.16 (1.46x)|

---

> > ### Comment · Reviewer_bAPx · 2025-08-04
> >
> > Given that inference is increasingly performed on NVIDIA H100 GPUs, the paper’s impact may be limited without explicit H100 support. Recent work on hardware acceleration (e.g., FA3) emphasizes the H100; adding H100 support would therefore strengthen the contribution. I encourage the authors to implement and evaluate their kernel(s) on the H100. Also the work targets inference rather than training, the compute cost should not that high.

---

> > > ### Author Response · Authors · 2025-08-08
> > >
> > > Dear reviewer bAPx,
> > >
> > > As the discussion period is coming to an end, we want to kindly ask if we were able to address your concerns regarding our method executed on H100. The Table above shows a consistent speedup.
> > >
> > > If there are other questions or concerns, feel free to ask. We are happy to answer them in the remaining time.
> > >
> > > Thank you!

---

> ### Author Response · Authors · 2025-08-05
>
> Per the request of the reviewer, we acquire a server with a h100 GPU and obtain some results using our current implementation as it is on h100. The setting is the same as table 2 mentioned in the paper and the A6000 results provided in the rebuttal. We can see that without any further optimization for h100 GPUs, we are already seeing a end-to-end speedup improvement for STree against Vanilla SD. This shows the potential of STree on better hardware. We will include more result with H100 GPUs in the future version of our paper.
>
>  | | MT_Bench |  | HumanEval|  | GSM8K | |
>  |:---|:---|:---|:---|:---|:---|:---|
>  |                 | Speed | $\tau$ | Speed|   $\tau$ | Speed | $\tau$ |
>  |                 |       | T=1 |       |      |       |      |
>  | Autoregressive  |76.66         |   1 | 77.67        |   1  |   78.35    |  1   |
>  | Vanilla SD      | 80.76 (1.05x)| 1.56| 85.63 (1.10x)| 1.64 | 87.04 (1.11x)| 1.63 |
>  | STree           | 85.18 (1.11x)| 2.04| 92.74 (1.19x)| 2.22 | 92.86 (1.19x)| 2.16 |

---

### Official Review · Reviewer_7Ar2 · 2025-07-03

**Clarity:** 2
**Significance:** 2
**Originality:** 3
**Rating:** 4
**Confidence:** 2

**Summary:**

This paper aims to accelerate state-space models (SSMs) using speculative decoding, a technique commonly employed in transformer-based large language models (LLMs). In speculative decoding, tree verification is often used to further enhance acceleration speed. However, due to fundamental differences between transformer-based LLMs and SSMs, the authors observe that token tree verification cannot be directly applied to SSMs accelerated with speculative decoding. Therefore, the authors propose STree, facilitating speculative tree decoding in SSMs.

**Questions:**

See above weaknesses and questions.

**Ethical Concerns:**

["NO or VERY MINOR ethics concerns only"]

**Final Justification:**

I have read the reply and will keep my current score.

**Limitations:**

yes

**Quality:**

3

**Strengths And Weaknesses:**

Strengths

1.	The motivation of this work is clear and meaningful. As the authors stated, it is necessary to facilitate speculative tree decoding in SSMs.

2.	The proposed method is natural and logically sound. The experimental results provide preliminary validation of its effectiveness.

Weaknesses and questions

1.	Speculative tree decoding has been explored for transformer-based LLMs, and speculative decoding has also been applied to SSMs. This work addresses a minor gap by proposing speculative tree decoding for SSMs; however, the challenges involved in this extension are limited, which reduces the novelty of the contribution.

2.	In line 26, the statement “AR Transformers leverage a sliding window of input tokens (context) as their ‘state’, which is ideal for speculative decoding since their context can be easily edited to remove unverified tokens” is unclear. Does this mean that AR Transformers retain all past tokens and use the attention mechanism to dynamically focus on them? The authors are encouraged to provide a clearer explanation.

3.	Code availability: I understand that the code will be released upon acceptance. However, NeurIPS allows and encourages authors to submit associated code files during the review process. This would help reviewers better assess the reproducibility of the work.

---

> ### Author Rebuttal · Authors · 2025-07-31
>
> We thank the reviewer for recognizing the motivation and methods of our work. We hereby address the concerns and questions raised by the reviewer:
>
> **W1: reduced novelty**: To the best of our knowledge, we are the first to apply speculative tree decoding to SSMs and hybrid models, which makes our work novel. While tree based speculative decoding exists for transformer based models, it heavily relies on the transformer's property that computing a tree of tokens doesn't incur much overhead with the use of attention masks. Without this property, it is much slower to compute a tree of tokens in SSMs due to their recurrent state update. Our work addresses this critical challenge by improving the efficiency of computing a tree of tokens with SSMs, which is not done in previous works.
>
> **W2: AR transformer clarification**: Yes, the reviewer is right that AR transformers retain the key and value of past tokens in memory (a structure called KV cache) to avoid recomputing them again for future generation. With speculative decoding, certain tokens computed might not be verified, which then needs to be removed from the KV cache to avoid contaminating future predictions. This can be done easily with KV cache, since it is partitioned on token boundaries and removing unverified tokens is equivalent to zeroing out the corresponding KV cache entries.
>
> **W3: Code availability**: We thank the reviewer for their interest in our work. Here we provide code segments where we apply the tree mask in the kernel to compute SSM output. We will publish the complete code base upon paper acceptance.
>
> ```
>     ...
>     # We don't need to iterate K since tree scan sequence length is probably short.
>     for k in range(0, K_MAX, BLOCK_SIZE_K):
>         cb = tl.load(cb_ptrs, mask=(offs_m[:, None] < chunk_size) & (offs_k[None, :] < chunk_size - k), other=0.0).to(tl.float32)
>         dA_cs_k = tl.load(dA_cumsum_ptrs, mask=offs_k < chunk_size - k, other=0.0).to(tl.float32)
>         # If there's seq_idx, we already set cb[i, j] = 0 for seq_idx[i] != seq_idx[j].
>         # So we don't need masking wrt seq_idx here.
>         cb *= tl.exp((dA_cs_m[:, None] - dA_cs_k[None, :]))
>         dt_k = tl.load(dt_ptrs, mask=offs_k < chunk_size - k, other=0.0).to(tl.float32)
>         cb *= dt_k
>
>         # apply tree masking here
>         tree_mask = tl.load(tree_mask_ptrs, mask=(offs_m[:, None] < chunk_size) & (offs_k[None, :] < chunk_size - k), other=0.0).to(tl.int1)
>         cb = tl.where(tree_mask, cb, 0)
>
>         cb = cb.to(x_ptr.dtype.element_ty)
>         x = tl.load(x_ptrs, mask=(offs_k[:, None] < chunk_size_limit - k) & (offs_n[None, :] < hdim), other=0.0)
>         acc += tl.dot(cb, x)
>         cb_ptrs += BLOCK_SIZE_K * stride_cb_csize_k
>         x_ptrs += BLOCK_SIZE_K * stride_x_seqlen
>         dt_ptrs += BLOCK_SIZE_K * stride_dt_csize
>         dA_cumsum_ptrs += BLOCK_SIZE_K * stride_dA_cs_csize
>         tree_mask_ptrs += BLOCK_SIZE_K * stride_tm_csize2
>         ...
> ```

---

> > ### Comment · Reviewer_7Ar2 · 2025-08-06
> >
> > Thanks for the authors' reply. I have read the reply. Since they have addressed my concern, I will keep my positive score.

---

> ### Comment · Area_Chair_Vv9s · 2025-08-05
> **Author-Reviewer Discussions**
>
> Dear Reviewer 7Ar2,
>
> Thanks for your review! The authors have provided their rebuttal. Please respond to the authors and update your review as appropriate.
>
> Thank you!
> AC

---

### Official Review · Reviewer_wh4v · 2025-07-03

**Clarity:** 3
**Significance:** 3
**Originality:** 2
**Rating:** 3
**Confidence:** 4

**Summary:**

This paper introduces STree, the first scalable algorithm that enables tree-based speculative decoding for SSMs and hybrid SSM-Transformer architectures. Existing speculative SSMs do not leverage tree-based verification due to inefficiencies in handling tree structures. The authors propose a hardware-aware implementation that constructs and utilizes a tree mask like prior works do, but specifically designed to allow efficient verification by accumulating state transition matrices through a novel formulation.

**Questions:**

1. Could adaptive or learned tree structures (e.g., from a policy model) significantly boost performance?

2. How well would STree perform on larger models like 30B or 70B?

**Ethical Concerns:**

["NO or VERY MINOR ethics concerns only"]

**Final Justification:**

I choose to maintain my current score.

**Limitations:**

1. STree has higher overhead for short sequences; benefits are only realized at sufficient tree depth or acceptance rate.
2. Efficiency relies on static tree structures; adapting to diverse prompt types is not explored.

**Paper Formatting Concerns:**

No.

**Quality:**

3

**Strengths And Weaknesses:**

Strengths:
1. Stree is the first to enable tree-based speculative decoding for SSMs, implying the potential of using SD in hybrid models
2. The paper presents clear derivations showing how diagonal SSM transition matrices can be aggregated for tree decoding.


Weaknesses:
1. Experiments only conducted on 8B models, which may not generalize to much larger LLMs
2. The model Relies on handcrafted or beam-based static trees; does not explore adaptive or learned trees.

---

> ### Author Rebuttal · Authors · 2025-07-31
>
> We thank the reviewer for recognizing the novelty in tree-base speculative decoding for SSMs. We address the concerns raised by the reviewer below:
>
> **W1: Larger LLMs**: It is challenging to find open-source, pretrained hybrid models suitable for speculative decoding that are bigger than 30B parameters. We explored several options:
>   - Nemotron-H-47B: A hybrid model released by Nvidia in April. The smaller Nemotron-H-8B model can potentially act as the draft model. However, the model does not support generation with KV cache/SSM states, making it impossible for us to use.
>   - Jamba-mini-50B: A hybrid model from AI21. We cannot find any suitable draft model that is small in size and uses the same tokenizer as the target model.
>
> If the reviewer has any specific open-source models that are suitable we would be happy to run experiments with them.
>
> Absent end-to-end results on larger models, we analyze the runtime (in ms) of a forward pass through Mamba2 models using different methods below.
>  |    | 2.7B | 7B | 13B | 23B |
>  |:---|:-------|:--------|:-------|:--------|
>  | Autoregressive |  10.95 |  22.94 | 40.69 | 72.41 |
>  | STree |   22.36 (2.04x) | 33.79 (1.47x) | 55.90 (1.37x) | 91.08 (1.26x) |
>  | Vanilla SD |  15.05 (1.37x) | 26.46 (1.15x) | 45.95 (1.13x) | 76.74 (1.06x) |
>
> Theoretically, the effectiveness of speculative decoding is determined by the runtime overhead of forward pass using different methods as well as the acceptance length. Both STree and Vanilla SD are paying an overhead (having a longer runtime than autoregressive forward pass) to compute multiple tokens. From the table below, we can see that the overhead of an STree forward pass is decreasing from 2.04x to 1.26x as we scale the model from 2.7B to 23B. The ratio between STree and Vanilla SD is also shrinking (from 1.48 to 1.19) as model size increases. This means that, with a larger model, it is much easier to achieve an end-to-end performance gain.
>
> Given that we already achieved end-to-end performance gain in 8B models as shown in Table 2 in the paper, there is a reason to believe that our performance with bigger models would be even better, assuming that the acceptance length is about the same.
>
> **W2: Dynamic tree**: Our method does not rely on static trees and is applicable to any tree generation strategy. We note that optimizing the tree structure is orthogonal to our work. Our work focuses on enabling efficient computation of token tree with SSMs. With a baseline static tree, we are already able to obtain runtime improvement, as shown in Table 2 in the paper. We also note that the beam search tree we used in Table 1 is not static, as the expansion of the tree depends on the joint probability and will vary from sample to sample. Here, we additionally show the end-to-end result for drafting with a beam search tree. We use greedy decoding with M = 3 and N = 4 (M, N explained in table 1). The results are as follow:
>
>  |    | Autoregressive | Vanilla SD | STree w. static tree | STree w. beam search tree|
>  |:---|:-------|:--------|:-------|:--------|
>  | Speed |  39.98 | 67.68 (1.69x) | 69.84 (1.75x) | 71.60 (1.79x) |
>  | $\tau$ |   1 | 2.04 | 2.47 | 2.60 |
>
> We can see that with a beam-search tree, we are able to get a better acceptance length and thus a faster end-to-end generation speed. We believe that more advance tree construction technique would bring even more benefits.

---

> ### Comment · Reviewer_wh4v · 2025-08-04
>
> Thanks for the authors' reply. The speedup performance of STree on larger models seems to be limited by its overhead. I choose to maintain my score.

---

> ### Author Response · Authors · 2025-08-04
>
> We thank the reviewer for the reply. I am afraid there might be some misunderstanding with our additional experiments provided in the rebuttal, resulting in an opposite conclusion with the reviewer. As we go from left to right across columns in the first table provide above, we can see that as model size increases, the overhead of STree relative to an autoregressive run decreases from 2.04x to 1.26x. This is a drastic drop in overhead. Meanwhile, if we compare it with Vanilla SD, the ratio between the overhead of STree and Vanilla SD also drops from 1.48 to 1.19. This means that given the same acceptance length achieved with draft models, a 23B model is able to have a higher inference speed-up than a 7B model using STree.
>
> We wonder if it is possible for the reviewer to elaborate more on why they think the speed up of STree on larger models seems to be limited by overhead. We would be happy to provide any further explanation.

---

> ### Author Response · Authors · 2025-08-08
>
> Dear reviewer wh4v,
>
> As the discussion period is coming to an end, we kindly ask you to review the Table above once more.
>
> The conclusion drawn from the Table is that our method has a **reduced** overhead at larger model size. It is not limited by the overhead, and our method **does scale** well to larger model sizes.
>
> We hope that this message gets to the Reviewer in time so that we can discuss this point of confusion.
>
> Thank you!

---

### Decision · Program_Chairs · 2025-09-17

**Decision:**

Accept (poster)

**Comment:**

This paper proposes STree, the first algorithm to perform tree-based speculative decoding in state-space models (SSMs) and hybrid architectures of SSMs and Transformer layers. The proposed method addresses the challenge of using SSMs to compute a token tree efficiently. The empirical evaluation demonstrates that STree outperforms vanilla autoregressive decoding and speculative decoding.

The reviewers generally believed the proposed method is technically sound, and the hardware-aware implementation will be practically useful for the community. Some reviewers asked about the scalability to larger models, and the authors provided additional experiments to address it. While a few reviewers still believed that the paper somewhat lacks novelty, they also regarded the contribution of the paper as satisfying the standard of NeurIPS. Reviewer wh4v's concerns, including the performance of STree on larger model sizes and the reliance on static trees, have been well-addressed by the authors during the rebuttal.

On balance, the AC recommends accepting the submission, as it could serve as a meaningful baseline for speculative decoding in SSMs.